# Is Erythrocyte Sedimentation Rate Necessary for the Initial Diagnosis of Giant Cell Arteritis?

**DOI:** 10.3390/life13030693

**Published:** 2023-03-03

**Authors:** Michael S. Hansen, Oliver N. Klefter, Lene Terslev, Mads R. Jensen, Jane M. Brittain, Uffe M. Døhn, Carsten Faber, Steffen Heegaard, Anne K. Wiencke, Yousif Subhi, Steffen Hamann

**Affiliations:** 1Department of Ophthalmology, Rigshospitalet, Faculty of Health and Medical Sciences, University of Copenhagen, DK-2100 Copenhagen, Denmark; 2Center for Rheumatology and Spine Diseases, Rigshospitalet, Faculty of Health and Medical Sciences, University of Copenhagen, DK-2100 Copenhagen, Denmark; 3Department of Clinical Physiology and Nuclear Medicine, Bispebjerg & Frederiksberg Hospital, DK-2400 Copenhagen, Denmark; 4Department of Clinical Physiology and Nuclear Medicine, Rigshospitalet, DK-2100 Copenhagen, Denmark; 5Eye Pathology Section, Department of Pathology, Rigshospitalet, DK-2100 Copenhagen, Denmark; 6Department of Clinical Research, University of Southern Denmark, DK-5230 Odense, Denmark

**Keywords:** giant cell arteritis, biomarkers, C-reactive protein, platelet count, erythrocyte sedimentation rate, diagnostic test accuracy, sequential biomarker analysis

## Abstract

Giant cell arteritis (GCA) is an ophthalmological emergency that can be difficult to diagnose and prompt treatment is vital. We investigated the sequential diagnostic value for patients with suspected GCA using three biochemical measures as they arrive to the clinician: first, platelet count, then C-reactive protein (CRP), and lastly, erythrocyte sedimentation rate (ESR). This retrospective cross-sectional study of consecutive patients with suspected GCA investigated platelet count, CRP, and ESR using diagnostic test accuracy statistics and odds ratios (ORs) in a sequential fashion. The diagnosis was established by experts at follow-up, considering clinical findings and tests including temporal artery biopsy. A total of 94 patients were included, of which 37 (40%) were diagnosed with GCA. Compared with those without GCA, patients with GCA had a higher platelet count (*p* < 0.001), CRP (*p* < 0.001), and ESR (*p* < 0.001). Platelet count demonstrated a low sensitivity (38%) and high specificity (88%); CRP, a high sensitivity (86%) and low specificity (56%); routine ESR, a high sensitivity (89%) and low specificity (47%); and age-adjusted ESR, a moderate sensitivity (65%) and moderate specificity (65%). Sequential analysis revealed that ESR did not provide additional value in evaluating risk of GCA. Initial biochemical evaluation can be based on platelet count and CRP, without waiting for ESR, which allows faster initial decision-making in GCA.

## 1. Introduction

Giant cell arteritis (GCA) is a systemic vasculitis which involves large- and medium-sized arteries [1]. The incidence of GCA is highest in Scandinavia [2]. A Swedish study estimated an annual incidence of 14 per 100,000 individuals aged ≥50 years [2]. Inflammation of the artery walls in GCA causes stenosis and occlusion, which may lead to visual loss due to ischemic optic neuropathy or central retinal artery occlusion (CRAO) [1]. Beyond affecting the eye, GCA can cause a stroke [1]. Thus, GCA is a medical and ophthalmological emergency, and prompt treatment is vital.

Clinically, symptoms of GCA can include headache, scalp tenderness, jaw claudication, fever, fatigue, loss of appetite, myalgia, and monocular transient or permanent loss of vision [1]. The gold standard in confirming the diagnosis of GCA has historically been a temporal artery biopsy (TAB) [3,4]. Treatment with high-dose corticosteroids is commenced upon clinical suspicion, as waiting days, or even hours, for the planning of TAB and subsequent histopathological confirmation of the diagnosis would put the patient at unnecessary risk of further ischemic complications. International consensus on the clinical practice and textbook information regarding GCA report that patients should be treated upon clinical suspicion [1,5,6,7,8]. However, symptoms can be subtle and may be a manifestation of other diseases; therefore, diagnosing GCA can be challenging. Further, symptoms alone have a low diagnostic accuracy for GCA [9].

Biochemical measures are of great importance in the diagnostic consideration of GCA [9]. These include markers of inflammation, i.e., platelet count, C-reactive protein (CRP), and erythrocyte sedimentation rate (ESR). Typically, blood differential count including platelet count is readily available within a few minutes. The results of the CRP are available within 15 to 30 min. Even faster results may be available using point-of-care C-reactive proteins. The ESR takes a longer time and is available at the earliest after 60 min. These delays pose interesting questions in clinical practice. First, in dubious cases where the suspicion of GCA is low but biochemical measures are taken to rule out GCA, is it worthwhile waiting 1 to 2 h for the ESR result? Or do we have enough information to make decisions earlier? Second, current studies are exploring intravenous alteplase in non-arteritic CRAO within 4 to 5 h after symptom onset [10,11]. In that context, evaluation of whether a CRAO is arteritic or non-arteritic needs to be performed as fast as possible, and one can question how much more value we obtain by waiting for the ESR result? Is it possible to obtain a biochemical indication of the diagnosis earlier?

To answer these important questions and to guide clinical practice, this study investigated the sequential value of biochemical measures (Figure 1), i.e., first, platelet count, then CRP, then ESR, in evaluating the risk of GCA in patients presenting relevant symptoms.

## 2. Materials and Methods

### 2.1. Study Design and Ethics

This was a retrospective cross-sectional study in a large tertiary center in Denmark. Ethical committee approval was obtained from the Ethics Committee for the Capital Region of Denmark (H-20082624). All aspects of this study adhered to the principles of the Declaration of Helsinki and to hospital guidelines for research conduct.

### 2.2. Patient Eligibility

In Denmark, all citizens have the right to access to specialized care, without out-of-pocket payment. This is financed through a tax-based healthcare coverage system. Upon symptoms, patients can contact their family physician or a primary care ophthalmologist during working hours, and an emergency telephone after working hours. These entities can refer to a hospital department for immediate emergency work-up. Therefore, upon arriving at the hospital, the personnel are already aware of the symptoms and can prioritize blood work-up and diagnostics accordingly in a timely manner. This organization allows an opportunity for timely diagnosis and commencement of treatment. The Danish health system also covers any diagnostic work-up necessary and the following management after the acute phase, including all necessary follow-ups. Medication at the pharmacy is paid for out-of-pocket but with a payment capping system that ensures affordability across the population. These circumstances also minimize any household income-based selection bias in our study, as any capacity for out-of-pocket payment or care dependency on insurance-coverage are irrelevant in the Danish healthcare system.

We included consecutive patients with suspected GCA and assessed between August 2018 and June 2019 at the Department of Ophthalmology and Department of Rheumatology and Spine Diseases at Rigshospitalet, Denmark. Patients were either evaluated by the on-call ophthalmologist because of ophthalmic symptoms or physicians from other departments in the absence of ophthalmic symptoms.

All patients underwent clinical examination and blood sampling for biochemical testing. Patients in whom GCA could not be ruled out commenced systemic glucocorticoid treatment and were referred for TAB. An ophthalmological examination was performed if ophthalmic symptoms were present, which included a minimum of best-corrected visual acuity, slit-lamp biomicroscopy, and fundus examination.

### 2.3. Biochemical Measurements

We measured complete blood count which included platelet count, CRP, and ESR. CRP was measured using Vitros 4600/5600 (Ortho Clinical Diagnostics, Raritan, NJ, USA). Measurements of platelet count were reported in 10^9^/L. The laboratory reference range defined <400 × 10^9^/L as low/normal and ≥400 × 10^9^/L as elevated. Measurements of CRP were reported in mg/L. The laboratory reference range defined <10 mg/L as normal and ≥10 mg/L as elevated. ESR was measured using the Sed Rate Screener 20/II (Greiner Bio-One GmbH, Kremsmünster, Austria). Measurements of ESR were reported in 1-h mm. The laboratory reference range defined normal as 2 to 20 mm in women and 2 to 15 mm in men. However, it is well-described that ESR increases with advanced age. Therefore, we also used an age- and gender-corrected normal ESR range, which was defined as: ≤age/2 for men and ≤(age + 10)/2 for women [12]. Platelet count was measured using a Sysmex XN-9000 (Sysmex Corporation, Kobe, Japan). Measurements of platelet count were in number of platelets per L. According to laboratory reference ranges, we defined <400 × 109/L as low/normal and ≥400 × 109/L as elevated. Definitions and ranges are summarized in Table 1.

### 2.4. Diagnosis of GCA

The American College of Rheumatology classification criteria were noted for all patients [3]. These criteria should be not considered diagnostic for GCA; instead, they are used in the clinic to help with diagnosis and differentiate GCA from other vasculitis types [4]. These criteria were: (i) age of onset ≥50 years, (ii) headache as a new symptom, (iii) temporal artery abnormality such as tenderness to palpation or decreased pulsation, (iv) ESR ≥50 mm, (v) abnormal TAB with features characteristic of GCA.

Experienced physicians in neuro-ophthalmology and rheumatology decided on the final clinical diagnosis at 6-months follow-up. A diagnosis of either definite GCA or definite non-GCA were given by reviewing the results of symptoms, examination findings, medication, blood biochemical measurements, and diagnostic tests which could be a combination of vascular ultrasound, 18-fluorodeoxyglucose positron emission tomography, and/or TAB. Uncertain cases were discussed between specialists until consensus [13].

### 2.5. Data Analysis and Statistics

Data were analyzed using IBM SPSS Statistics version 28.0.1.0 (IBM, Armonk, NY, USA) and MedCalc (MedCalc Software Ltd., Ostend, Belgium). Categorical data were summarized in numbers and percentages and compared using the χ^2^-test. Continuous variables were evaluated visually for normal distribution, and if normal distribution was present, such data were reported using mean and standard deviation and compared using parametric tests. Continuous variables without normal distribution were reported using the median and interquartile range (IQR) and compared using non-parametric tests.

For diagnostic test accuracy measures, results were reported in a 2 × 2 contingency table. For the three biochemical measures investigated, we calculated sensitivity, specificity, the area under the curve (AUC) of the receiver operating characteristic (ROC) curve, the positive predictive value, and the negative predictive value. Odds ratios (ORs) were used to calculate the association with a later diagnosis of GCA in a sequential fashion representative of the time of arrival of biochemical measures (platelet count: <15 min, CRP: 15 to 30 min, ESR: 60 to 120 min). Where possible, 95% confidence intervals (95% CI) are reported. *p*-values <0.05 were interpreted as statistically significant.

## 3. Results

### 3.1. Study Patients

A total of 106 consecutive patients with suspected GCA were enrolled in this study. Of these, 12 were excluded due to incomplete biochemical data for any reason. Hence, a total of 94 patients were included for analyses, of which 37 (40%) were later diagnosed with GCA. Patients presented with ocular symptoms in 46 (49%) cases. Ocular symptoms were categorized as vision loss (n = 16), blurry vision (n = 10), diplopia (n = 9), amaurosis fugax (n = 5), ocular pain (n = 4), and transient diplopia (n = 2). Patients with GCA did not differ significantly in demographic data from those without GCA (Table 2). Clinical characteristics using data obtained later, after the TAB, are also summarized in Table 2.

### 3.2. Platelet Count, CRP, and ESR between Patients with and without GCA

Patients with GCA, when compared with those without GCA, had a significantly higher platelet count (median 263 × 10^9^/L vs. 370 × 10^9^/L, *p* < 0.001, Mann–Whitney U test), CRP (median 8.0 mg/L vs. 41.0 mg/L, *p* < 0.001, Mann–Whitney U test), and ESR (median 21 mm vs. 58 mm, *p* < 0.001, Mann–Whitney U test). These differences remained significant after categorizing biochemical measures according to whether these values were elevated according to their standard biochemical range values (Table 3).

### 3.3. Diagnostic Test Accuracy of Platelet Count, CRP, and ESR for the Detection of GCA

We compared the diagnostic test accuracy of platelet count, CRP, and ESR using standard laboratory references and age-adjusted references for the detection of GCA.

The diagnostic test accuracy statistics of the platelet count showed a low sensitivity (38%, 95% CI: 22 to 55%), a high specificity (88%, 76 to 95%), and a significant discriminatory ability (AUC: 0.628; 95% CI: 0.508 to 0.747; *p* = 0.04).

The diagnostic test accuracy statistics of CRP differed and showed a high sensitivity (86%, 95% CI: 71 to 95%), a low specificity (56%, 95% CI: 42 to 69%), and a significant discriminatory ability as well (AUC: 0.713; 95% CI: 0.608 to 0.818; *p* = 0.001).

The diagnostic test accuracy of the standard reference ESR showed a high sensitivity (89%, 74 to 97%), a low specificity (47%, 34 to 61%), and a significant discriminatory ability (AUC: 0.683; 95% CI: 0.575 to 0.790; *p* = 0.003).

The diagnostic test accuracy of the age-adjusted ESR was moderately sensitive (65%, 47 to 80%) and moderately specific (65%, 51 to 77%) and demonstrated a significant discriminatory ability (AUC: 0.649; 95% CI: 0.534 to 0.764; *p* = 0.015).

Further details of the diagnostic test accuracy and statistics of these biochemical measurements are outlined in Table 4.

### 3.4. Likelihood of GCA Based on Sequential Evaluation of Platelet Count, CRP, and ESR

We first considered the likelihood of GCA based on whether the platelet count was <400 × 109/L or ≥400 × 109/L. An elevated platelet count led to a significantly higher likelihood of GCA (OR: 4.35; 95% CI: 1.55 to 12.22; *p* = 0.005).

In the next step, we considered whether the CRP value was <10 mg/L or ≥10 mg/L and its impact on the sequential likelihood of GCA. Among those with an initial low or normal platelet count, an elevated CRP led to a significantly higher likelihood of GCA (OR: 7.13; 95% CI: 2.11 to 24.08; *p* = 0.002). However, in those with an elevated platelet count, an elevated CRP did not provide further value to the likelihood of GCA (OR: 5.20; 95% CI: 0.38 to 70.91; *p* = 0.2). In patients with GCA with a low/normal CRP (n = 5), the CRP was measured prior to commencement of corticosteroid treatment in four of five patients (80%).

In the final step, we considered the value of an elevated ESR on the sequential likelihood of GCA. Among those with an elevated platelet count and an elevated CRP (n = 18), an elevated ESR was not significantly associated with a diagnosis of GCA (standard reference: OR 2.45; 95% CI: 0.04 to 139.96; *p* = 0.7; age-adjusted reference: OR: 2.39; 95% CI: 0.10 to 58.78; *p* = 0.6). Among those with a normal platelet count and a normal CRP (n = 34), an elevated ESR was not significantly associated with a diagnosis of GCA (standard reference: OR: 3.29; 95% CI: 0.39 to 27.78; *p* = 0.3; age-adjusted reference: OR: 3.00; 95% CI: 0.23 to 38.74; *p* = 0.4). In other words, when the platelet count and the CRP were in interpretive agreement, the ESR—regardless of using standard references or age-adjusted references—did not provide any further diagnostic value to the sequential likelihood of GCA.

We then looked at cases with inconsistent results from platelet count and CRP. Among those with an elevated platelet count and a normal CRP (n = 3), one individual had GCA, and the ESR was elevated according to the reference values in all patients and particularly so in two patients—of which one was the patient with GCA. In this scenario, an elevated ESR was not significantly associated with a later diagnosis of GCA (standard reference: OR: 0.33; 95% CI: 0.01 to 16.80; *p* = 0.6; age-adjusted reference: OR: 3.00; 95% CI: 0.06 to 151.20; *p* = 0.6). Among those with a normal platelet count and an elevated CRP (n = 39), an elevated ESR was not significantly associated with a later diagnosis of GCA (standard reference: OR: 2.13; 95% CI: 0.34 to 13.24; *p* = 0.4; age-adjusted reference: OR: 1.13; 95% CI: 0.32 to 3.99; *p* = 0.9). Thus, in cases of inconsistent biochemical results, ESR did not provide any further diagnostic value to the sequential likelihood of GCA.

## 4. Discussion

In this study, we examined 94 patients with suspected GCA of whom 40% were later confirmed with a definite diagnosis of GCA. In this relatively large sample, we evaluated the sequential value of biochemical measures in evaluating the risk of GCA in patients presenting relevant symptoms. Platelet count, CRP, and ESR all showed significant discriminatory ability, and all were significantly higher in patients with GCA compared to those without. In a diagnostic test accuracy meta-analysis of laboratory tests for GCA, van der Geest et al. reported that platelet count exhibited a low sensitivity (45.8%) and high specificity (87.8%), that CRP exhibited a high sensitivity (87.4%) and low specificity (31.4%), and that ESR exhibited a high sensitivity (82.6 to 93.2% depending on definitions of >20 mm/h to >60 mm/h) and low specificity (33.8 to 70.5% depending on definitions of >20 mm/h to >60 mm/h) [9]. These diagnostic test accuracy statistics are similar to those seen in our study sample and highlight both the importance of prioritizing early blood sampling in cases of suspected GCA to facilitate early initial diagnosis and also that biochemical measurements only provide a moderate level of diagnostic input.

When evaluating the likelihood of GCA and looking at values sequentially as they arrive, i.e., first, platelet count, then CRP, and then ESR, we found that clinical decisions can be made without the ESR. An elevated platelet count was associated with a high risk of GCA regardless of the results of CRP or ESR. A normal platelet count and an elevated CRP was associated with a high risk of GCA regardless of the results of the ESR. The results of ESR did not change the likelihood of GCA or any decision made using the combination of platelet count and CRP. These results give rise to crucial considerations when pursuing a faster initial diagnosis for dubious cases to facilitate prompt glucocorticoid treatment for an atypical clinical presentation of GCA or treatment for relevant differential diagnoses e.g., non-arteritic CRAO and the ongoing studies of time-critical intravenous alteplase [10,11]. It should be noted that ultrasound examination of the extra-cranial arteries may enable a rapid and immediate diagnosis of GCA in a rheumatological setting [14,15]. However, ultrasound examination is not readily available in all clinics; hence, the current study adds valuable information to clinical decision making.

The exact pathophysiology of GCA remains to be fully elucidated, but inflammation develops in the artery wall with the recruitment of T-cells and monocytes, which transform into macrophages and eventually into the so-called giant cells [16]. Macrophages and T-cells within these vasculitis lesions secrete a spectrum of proinflammatory cytokines which importantly include interleukin-1 beta and interleukin-6 [16,17]. These cytokines induce a systemic acute-phase response which includes hepatic CRP secretion; thus, systemic CRP mirrors systemic inflammation and there is a clear correlation between the severity of the inflammation and the systemic levels of CRP [18]. The vascular inflammation leads to occlusion of the lumen, which leads to ischemia and the symptoms classically seen in the disease [16]. The systemic inflammation contributes to general illness and symptoms such as fever, fatigue, and loss of appetite [16]. The systemic inflammation also promotes thrombocytosis and tends to clump erythrocytes, which leads to a higher sedimentation rate of erythrocytes and therefore an increased ESR.

The severity of lumen occlusion and symptoms may vary, which can make clinical diagnosis difficult. Patients with suspected GCA are often elderly individuals, and elderly individuals can also present with numerous other causes for visual impairment. Together with other more benign causes of muscular tenderness and/or headache, these cases can be clinically difficult to distinguish from GCA without an ophthalmological examination. One highly prevalent ocular co-morbidity among the elderly is age-related macular degeneration (AMD) [19,20], which is associated with changes in systemic immunity [21,22,23,24]. Specifically, the development of choroidal neovascularization, which can lead to a significant loss of central vision, is associated with flares of the systemic immune system [25]. Patients with AMD have generally higher levels of systemic CRP [24,26]. It has also been reported that patients with AMD have a higher neutrophil-to-lymphocyte-ratio, which suggests alteration of the differential cell counts [23]. However, all systemic changes found in patients with AMD are subtle and thus clearly distinguishable from the clear immunological activity observed in patients with GCA. Similar subtle immunological findings are observed for other prevalent causes of vision loss in the elderly [27,28,29,30]. These aspects demonstrate the challenges in diagnosing GCA in the elderly, as they can have various other co-morbidities.

Strengths and limitations need to be acknowledged when interpreting the results of this study. First, this is a retrospective study of individuals seen in the clinic. Thus, there is a selection bias, as those without any suspicion of GCA are ruled out based on symptoms. However, this was a large study of patients evaluated consecutively in a tertiary center throughout one year, which highlights a strength to minimize selection bias. Further, due to the retrospective nature of the study, systematic data collection on all diagnostic aspects was not performed; hence, it is difficult to obtain a full picture of the final diagnoses of individuals without GCA. Second, we used the American College of Rheumatology 1990 classification criteria [3]. The American College of Rheumatology updated their classification criteria in 2022, which were not used in this study [31]. Third, we evaluated both standard-range ESR and age-adjusted ESR in this study. Interestingly, standard-range ESR provided a diagnostic test accuracy similar to that of CRP, whereas the age-adjusted ESR provided a moderate sensitivity and specificity profile. It should be noted that the age-based increase in the ESR reference is based on a study based on individuals up to the age of 65 years [12], which challenges its validity in a GCA population which consists of significantly older patients, e.g., in our study, patients were aged 74.1 ± 7.6 years. Fourth, in interpreting the results of our study, it should be noted that levels of inflammatory markers are also influenced by the extent and subtype of GCA and of any co-morbidities [32]. Finally, the final diagnosis of GCA is based on an evaluation of the complete clinical and biochemical picture. Therefore, the diagnostic classification of GCA is, at least theoretically, partly based on the results of ESR. This introduces a bias when estimating the value of ESR for the diagnosis of GCA.

It remains important that cases with a strong clinical suspicion of GCA commence relevant treatment upon suspicion. Presently, there is no single laboratory test, symptom, or examination that can perfectly diagnose or rule out GCA; the diagnosis of GCA is therefore complex and relies on a comprehensive picture of the patient. However, in cases with a low suspicion of GCA, we argue that initial biochemical evaluation can be made based on platelet count and CRP without waiting for the ESR result. In particular, it should be noted that the platelet count provides a high specificity, and that CRP provides a high sensitivity. This strategy allows for faster initial decision-making regarding whether the patient should start corticosteroid treatment or whether other important differential diagnoses requiring various time-critical treatments should be considered. Finally, these considerations follow the general notion of clinical practice in many centers around the world in which ESR is becoming an increasingly obsolete biochemical measure for the diagnosis of GCA.

## Figures and Tables

**Figure 1 life-13-00693-f001:**
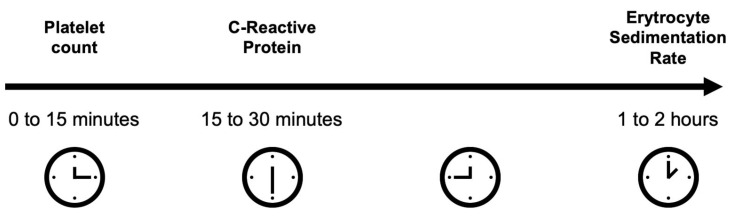
In routine clinical practice, the time before biochemical results can be obtained varies significantly between platelet count, C-reactive protein, and erythrocyte sedimentation rate.

**Table 1 life-13-00693-t001:** Laboratory range definitions.

	Low/Normal	Elevated
Platelet count, 10^9^/L	<400	≥400
CRP, mg/L	<10	≥10
ESR (routine), mm	≤15 mm in men ≤20 mm in females	>15 mm in men >20 mm in females
ESR (age-adjusted), mm	≤age/2 for men ≤(age + 10)/2 for women	>age/2 for men >(age + 10)/2 for women

Abbreviations: CRP = C-reactive protein; ESR = erythrocyte sedimentation rate.

**Table 2 life-13-00693-t002:** Patient demographics and clinical characteristics.

	Patients without GCA (n = 57)	Patients with GCA (n = 37)	*p*-Value
Age, years, mean ± SD	72.9 ± 10.7	74.1 ± 7.6	0.5
Gender, n (%)			0.2
Females	35 (61)	27 (73)	
Males	22 (39)	10 (27)	
ACR criteria, n (%) ^a^			<0.001
≥3	15 (26)	31 (84)	
<3	42 (74)	6 (16)	
TAB, n (%) ^b^			<0.001
Positive	0 (0)	23 (68)	
Negative	54 (100)	11 (32)	

Age is compared using the independent samples *t*-test. The remaining variables are categoric and compared using the χ^2^-test. Abbreviations: ACR = American College of Rheumatology; GCA = giant cell arteritis; n = number (of patients); SD = standard deviation; TAB = temporal artery biopsy. ^a^: The ACR (1990) GCA classification criteria are: (i) age of onset ≥50 years; (ii) headache as a new symptom; (iii) temporal artery abnormality such as tenderness to palpation or decreased pulsation; (iv) ESR ≥50 mm; (v) abnormal TAB with features characteristic of GCA. ^b^: Valid data on TAB were not present in 6 cases (biopsy insufficient or incorrect, e.g., vein or nerve).

**Table 3 life-13-00693-t003:** Comparison of biochemical parameters between patients with and without giant cell arteritis.

	Patients without GCA (n = 57)	Patients with GCA (n = 37)	*p*-Value
Platelet count			
Median (IQR), 10^9^/L	263 (210 to 339)	370 (294 to 442)	<0.001
Low/Normal (<400 × 10^9^/L), n	50	23	
Elevated (≥400 × 10^9^/L), n	7	14	0.004
CRP			
Median (IQR), mg/L	8.0 (2.0 to 40.0)	41.0 (17.0 to 92.0)	<0.001
Low/Normal (<10 mg/L), n	32	5	
Elevated (≥10 mg/L), n	25	32	0.004
ESR			
Median (IQR) mm	21 (9 to 54)	58 (34 to 82)	<0.001
Low/Normal, n	27	4	
Elevated, n	30	33	<0.001
Age-adjusted ESR			
Low/Normal, n	37	13	
Elevated, n	20	24	0.005

Continuous variables are compared using the Mann–Whitney U test. Categoric variables are compared using the χ^2^-test. Abbreviations: CRP = C-reactive protein; ESR = erythrocyte sedimentation rate; GCA = giant cell arteritis; IQR = interquartile range; n = number (of patients).

**Table 4 life-13-00693-t004:** Diagnostic test accuracy of biochemical parameters between patients with and without giant cell arteritis.

		Patients without GCA (n = 57)	Patients with GCA (n = 37)	Performance (95% CI)
Platelet count ≥400 × 10^9^/L	Positive	7	14	Sensitivity: 38% (22 to 55%) Specificity: 88% (76 to 95%) AUC: 0.628 (0.508 to 0.747) PPV: 67% (47 to 82%) NPV: 68% (62 to 74%)
	Negative	50	23	
CRP ≥10 mg/L	Positive	25	32	Sensitivity: 86% (71 to 95%) Specificity: 56% (42 to 69%) AUC: 0.713 (0.608 to 0.818) PPV: 56% (48 to 64%) NPV: 86% (73 to 94%)
	Negative	32	5	
ESR >15 mm in men >20 mm in females	Positive	30	33	Sensitivity: 89% (74 to 97%) Specificity: 47% (34 to 61%) AUC: 0.683 (0.575 to 0.790) PPV: 52% (46 to 59%) NPV: 87% (72 to 95%)
	Negative	27	4	
Adjusted ESR	Positive	20	24	Sensitivity: 65% (47 to 80%) Specificity: 65% (51 to 77%) AUC: 0.649 (0.534 to 0.764) PPV: 55% (44 to 65%) NPV: 74% (64 to 82%)
	Negative	37	13	

A low/normal age-adjusted ESR range was defined as: ≤age/2 for men and ≤(age + 10)/2 for women.Abbreviations: AUC = area under the curve; CI = confidence interval; CRP = C-reactive protein; ESR = erythrocyte sedimentation rate; GCA = giant cell arteritis; PPV = positive predictive value; n = number (of patients); NPV = negative predictive value.

## Data Availability

The data presented in this study are not publicly available due to patient confidentiality and the permissions upon which ethical approval was given for this study.

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
