# Peer review of "Is Erythrocyte Sedimentation Rate Necessary for the Initial Diagnosis of Giant Cell Arteritis?"

_life, 2023, doi:10.3390/life13030693_

Round 1

Reviewer 1 Report

Thank you for allowing me to review this manuscript. It is well written and I have only a few comments for consideration.

While I like the concept of figure 1, I think the hands on the stop watch or clock need to change to make it more impactful. 

Paragraph 2.2 lines 102-115 could be reduced to 1-2 sentences

Table 1, why is the ESR routine in bold?

Please note somewhere within the document that the ACR have been revised but at the time of this study you classified patients retrospectively with this tool (line 147ish)

Ponte C, Grayson PC, Robson JC, Suppiah R, Gribbons KB, Judge A, Craven A, Khalid S, Hutchings A, Watts RA, Merkel PA, Luqmani RA; DCVAS Study Group. 2022 American College of Rheumatology/EULAR classification criteria for giant cell arteritis. Ann Rheum Dis. 2022 Dec;81(12):1647-1653. doi: 10.1136/ard-2022-223480.

Can you clarify what valid data on TAB meant ? incorrect or not enough specimen? Line 201

Line 319 should it say increased ESR?

I think the paragraph on AMD is not relevant and could be removed, its not really a contender for a differential in GCA. Lines 325-336

Please comment on cranial disease having lower inflammatory markers as compared to those with systemic GCA or LV GCA

I would also consider what future serum biomarkers could be used or help?? IL-6 measurements? VEGF??

Author Response

Reviewer #1 general comments:

Thank you for allowing me to review this manuscript. It is well written and I have only a few comments for consideration.

Authors’ response:

Thank you for your time and comments.

Reviewer #1 comment #1:

While I like the concept of figure 1, I think the hands on the stop watch or clock need to change to make it more impactful.

Authors’ response:

The figure is now revised as recommended.

Reviewer #1 comment #2:

Paragraph 2.2 lines 102-115 could be reduced to 1-2 sentences

Authors’ response:

Thank you for this comment, which we have taken into consideration. As highlighted by another reviewer, it may be of stronger benefit to leave the paragraph lengthy as it may better the understanding of the context in which our study has been conducted.

Reviewer #1 comment #3:

Table 1, why is the ESR routine in bold?

Authors’ response:

This seems to be a formatting issue. It is now fixed.

Reviewer #1 comment #4:

Please note somewhere within the document that the ACR have been revised but at the time of this study you classified patients retrospectively with this tool (line 147ish)

Ponte C, Grayson PC, Robson JC, Suppiah R, Gribbons KB, Judge A, Craven A, Khalid S, Hutchings A, Watts RA, Merkel PA, Luqmani RA; DCVAS Study Group. 2022 American College of Rheumatology/EULAR classification criteria for giant cell arteritis. Ann Rheum Dis. 2022 Dec;81(12):1647-1653. doi: 10.1136/ard-2022-223480.

Authors’ response:

Thank you for this suggestion. We have now added this consideration in our discussion.

Reviewer #1 comment #5:

Can you clarify what valid data on TAB meant ? incorrect or not enough specimen? Line 201

Authors’ response:

Invalid data indicates biopsies that were incorrect (we had examples of veins and nerves) or insufficient/not enough. We have now clarified this in Table 2.

Reviewer #1 comment #6:

Line 319 should it say increased ESR?

Authors’ response:

Thank you for pointing this out. We agree and have revised as suggested.

Reviewer #1 comment #7:

I think the paragraph on AMD is not relevant and could be removed, its not really a contender for a differential in GCA. Lines 325-336

Authors’ response:

Thank you for this comment, which we have taken into consideration. Presumably because of the age overlap, many of our patients with GCA also has various degrees of AMD. Considering the extensive reports on low-grade inflammation in AMD and the potential influence of the immune system on AMD, we understand that this is a topic that needs to be touched upon as a contrast to that of GCA, since we completely agree with the reviewer that clinics and the biochemical response in GCA is far from that of AMD. Shortening paragraphs to make points brief is a good suggestion but contradicts with previous editorial correspondences with us asking for a more lengthy discussion of our points.

Reviewer #1 comment #8:

Please comment on cranial disease having lower inflammatory markers as compared to those with systemic GCA or LV GCA

Authors’ response:

Thank you for this suggestion. We have now included this comment in our discussion.

Reviewer #1 comment #9:

I would also consider what future serum biomarkers could be used or help?? IL-6 measurements? VEGF?? 

Authors’ response:

Thank you for these suggestions. Although it would be interesting to speculate on the biomarkers of future, we want to keep the focus of this paper on the value of existing biomarkers in the timely diagnosis of GCA.

Reviewer 2 Report

The authors investigated the sequential diagnostic value for patients with suspected GCA using three biochemical measures: Platelet count, C reactive protein and erythrocyte sedimentation rate. The manuscript is well written and the results are clearly reported. The topic is very interesting and can add a significant value to the diagnostic field of the GCA. 

Author Response

Reviewer #2 general comments:

The authors investigated the sequential diagnostic value for patients with suspected GCA using three biochemical measures: Platelet count, C reactive protein and erythrocyte sedimentation rate. The manuscript is well written and the results are clearly reported. The topic is very interesting and can add a significant value to the diagnostic field of the GCA.

Authors’ response:

Thank you for your time and comments.

Reviewer 3 Report

General comment : This is a very good paper on the potential usefulness/uselessness of measuring erythrocyte sedimentation rate (ESR) in the workup of potential giant-cell arteritis (GCA). The authors retrospectively explored a blood test sequence of platelet counts, CRP and ESR, for their contribution to the final diagnosis of GCA, vs other diagnoses. Globally, I agree with their conclusion that ESR is of little added-value for the final clinical diagnosis of GCA, as already reported in the literature. The paper is well written and scientifically sound.

Specific comments:

-          The authors used the standard ACR criteria back in 1990, which is fair. However, ESR is one of the 5 criteria for guiding to the diagnosis, when 3 out of 5 are required. Challenging the value of ESR thereafter is a bit odd, as it part of the final diagnosis, on the basis of the ACR criteria. The authors should address the following issue: what would happen if they do not include ESR in the ‘final diagnosis’, by using rather the “Tree format” of the 1990 ACR paper that excludes ESR from the diagnosis (and add jaw or tongue claudication and scalp tenderness as additional criteria). From a methodological viewpoint, this would eliminate the risk of run-in bias.

-          It is not clear what ESR value is used. The authors report ESR (as usual in the literature, and wrongly), as mm/hour. Actually, they should -in this particular study- express it as ESR after the first hour or after two hours, since this measure is not linearly correlated over time. Ideally, if 1-hr ESR is available, this should be the only data to report.

-          The authors rightly point out that wasting time for biochemical evaluations should be avoided. According to the point before, the difference between the time delay to get a platelet count or a CRP level, remains within an hour, which makes the gain over time very marginal. In addition, it would be important to know the time between the initiation of symptoms and consultation: if this time lapse is in the order of 24-72 hr, what is the actual benefit of gaining half an hour for biochemical clues.

-          For clarity, the diagnoses in patients with no GCA should be mentioned briefly, I assume dozens shall be reported!

-          The discussion section should be rewritten according to potential replies of the aforementioned points.

-          I like very much the description of the Danish Healthcare system, which is an information that is most often lacking in papers and I want to congratulate the authors for putting this in writing.

Minor comments:

-          In section 3.1, the authors refer to Table 1, when it is Table 2 (line 8) and on the next line again refer to Table 2. Please correct.

-          In Table 3, there is an inversion of data for ESR, where there should be 33 elevated and 4 normal in the GCA group!

References: 2nd line 2.4, the reference (4) is not correct, it should be (3), please check the whole manuscript about this.

Author Response

Reviewer #3 general comments:

General comment : This is a very good paper on the potential usefulness/uselessness of measuring erythrocyte sedimentation rate (ESR) in the workup of potential giant-cell arteritis (GCA). The authors retrospectively explored a blood test sequence of platelet counts, CRP and ESR, for their contribution to the final diagnosis of GCA, vs other diagnoses. Globally, I agree with their conclusion that ESR is of little added-value for the final clinical diagnosis of GCA, as already reported in the literature. The paper is well written and scientifically sound.

Authors’ response:

Thank you for your time and comments.

Reviewer #3 comment #1:

-          The authors used the standard ACR criteria back in 1990, which is fair. However, ESR is one of the 5 criteria for guiding to the diagnosis, when 3 out of 5 are required. Challenging the value of ESR thereafter is a bit odd, as it part of the final diagnosis, on the basis of the ACR criteria. The authors should address the following issue: what would happen if they do not include ESR in the ‘final diagnosis’, by using rather the “Tree format” of the 1990 ACR paper that excludes ESR from the diagnosis (and add jaw or tongue claudication and scalp tenderness as additional criteria). From a methodological viewpoint, this would eliminate the risk of run-in bias.

Authors’ response:

Thank you for this comment. This is an important limitation of our study which needs to be acknowledged. Therefore, it is now included in our discussion of study limitations.

Reviewer #3 comment #2:

-          It is not clear what ESR value is used. The authors report ESR (as usual in the literature, and wrongly), as mm/hour. Actually, they should -in this particular study- express it as ESR after the first hour or after two hours, since this measure is not linearly correlated over time. Ideally, if 1-hr ESR is available, this should be the only data to report.

Authors’ response:

Thank you for this point. We used the 1-hr ESR. This is now specified in the manuscript as recommended.

Reviewer #3 comment #3:

-          The authors rightly point out that wasting time for biochemical evaluations should be avoided. According to the point before, the difference between the time delay to get a platelet count or a CRP level, remains within an hour, which makes the gain over time very marginal. In addition, it would be important to know the time between the initiation of symptoms and consultation: if this time lapse is in the order of 24-72 hr, what is the actual benefit of gaining half an hour for biochemical clues.

Authors’ response:

Thank you for raising this important point. One aspect is that timely commencement of treatment for GCA is important, although few hours may have small if any value. However, another and perhaps more important point is that ongoing RCT studies for thrombolysis of CRAO (within 4.5 hours of visual onset) need to rule out GCA. In this context, a difference of 1 hour is huge. We have now revised to better clarity on this point in our introduction.

Reviewer #3 comment #4:

-          For clarity, the diagnoses in patients with no GCA should be mentioned briefly, I assume dozens shall be reported!

Authors’ response:

Thank you for this comment. We agree that patients without GCA still present with a range of findings that need to be evaluated, diagnosed, and treated as appropriate. We have listed ocular symptoms in lines 205–207. The full picture of the diagnoses in patients without GCA is a complex picture, and our retrospective study design and the ethical approval for which it is based does not allow listing of the full picture of such diagnoses. A prospective study would be needed to collect relevant information for this purpose. We acknowledge this as a limitation in the discussion and have now elaborated on this topic.

Reviewer #3 comment #5:

-          The discussion section should be rewritten according to potential replies of the aforementioned points.

Authors’ response:

We have revised the discussion section according to the important points made by the reviewer.

Reviewer #3 comment #6:

-          I like very much the description of the Danish Healthcare system, which is an information that is most often lacking in papers and I want to congratulate the authors for putting this in writing.

Authors’ response:

Thank you.

Reviewer #3 comment #7:

-          In section 3.1, the authors refer to Table 1, when it is Table 2 (line 8) and on the next line again refer to Table 2. Please correct.

Authors’ response:

This is now corrected.

Reviewer #3 comment #8:

-          In Table 3, there is an inversion of data for ESR, where there should be 33 elevated and 4 normal in the GCA group!

Authors’ response:

Thank you for pointing this out, which is now corrected.

Reviewer #3 comment #9:

References: 2nd line 2.4, the reference (4) is not correct, it should be (3), please check the whole manuscript about this.

Authors’ response:

This is now corrected.

Round 2

Reviewer 3 Report

The authors have adequately revised the manuscript and addressed the issues of the initial version.